# Tensile and Flexural Behaviors of Basalt Textile Reinforced Sprayed Glass Fiber Mortar Composites

**DOI:** 10.3390/ma16124251

**Published:** 2023-06-08

**Authors:** Ali Osman Ates, Gökhan Durmuş, Alper Ilki

**Affiliations:** 1Department of Civil Engineering, Faculty of Technology, Gazi University, Ankara 06560, Türkiye; aliates@gazi.edu.tr; 2Department of Civil Engineering, Faculty of Civil Engineering, Istanbul Technical University, Istanbul 34469, Türkiye; ailki@itu.edu.tr

**Keywords:** strengthening, material behavior, tensile characterization, test procedure, textile-reinforced mortar, fabric-reinforced cementitious matrix

## Abstract

The proposed study combines sprayed glass fiber-reinforced mortar and basalt textile-reinforcement to harness the favorable properties of each component to obtain a composite material that can be used for strengthening of existing structures. This includes crack resistance and a bridging effect of glass fiber-reinforced mortar and the strength provided by the basalt mesh. In terms of weight, mortars containing two different glass fiber ratios (3.5% and 5%) were designed, and tensile and flexural tests were conducted on these mortar configurations. Moreover, the tensile and flexural tests were performed on the composite configurations containing one, two, and three layers of basalt fiber textile reinforcement in addition to 3.5% glass fiber. Maximum stress, cracked and uncracked modulus of elasticity, failure mode, and average tensile stress curve results were compared to determine each system’s mechanical parameters. When the glass fiber content increased from 3.5% to 5%, the composite system without basalt textiles’ tensile behavior slightly improved. The increase in tensile strength of composite configurations with one, two, and three layers of basalt textile reinforcement was 28%, 21%, and 49%, respectively. As the number of basalt textile reinforcements increased, the slope of the hardening part of the curve after cracking clearly increased. Parallel to the tensile tests, four-point bending tests showed that the composite’s flexural strength and deformation capacities increase as the number of basalt textile reinforcement layers increase from one to two.

## 1. Introduction

Today, the retrofit of substandard or damaged structures or buildings using natural or artificial fibers is progressing rapidly all over the world, as well as in Turkey. According to the U.S. Army Corps of Engineers, concrete fiber reinforcement is defined as a composite material with randomly distributed fibers in concrete using fine or coarse aggregate [1]. Reinforcing fibers are produced in many types (steel, plastic, glass, nylon, polyester, polypropylene, etc.) and sizes, and with artificial and natural materials (cotton, hemp, and asbestos) [2]. Additionally, the durability of the fibers alters as a result of changes in their length and flexibility [3]. Especially basalt fibers, which contain 80% plagioclase and pyroxene, are preferred because they are natural and environmentally friendly [4]. Moreover, the ductility increase after addition of these fibers into the mixture of high [5] and normal strength concretes [6] or mortars is one of the most interesting reinforcement methods due to their very lightweight nature.

It is known that basalt fibers are produced by melting basalt rock at temperatures between 1500 and 1700 °C [7], homogenizing it without adding any additional chemicals [8], and offering a very cost-effective application area compared to other fibers in terms of environmental impact [9]. In addition, it is well known that the raw material cost and quality of these fibers depend on the production procedures [10]. Cement-based composites consisting of basalt textile-reinforced glass fiber mortar [11], textile-reinforced mortar (TRM), fabric-reinforced cementitious matrix (FRCM), or textile-reinforced concrete (TRC) are used for retrofitting of sub-standard concrete columns by external jacketing to enhance the stress–strain relationship [12].

For tensile performance, the stress–strain relationship of cement-based composite systems during load transfer is the most significant requirement [13]. It is known that fabric-reinforced systems are widely used in wall- and column-strengthening systems [14], columns that are subjected to vertical loading [15,16,17,18], in-plane behavior of walls under load [19,20,21,22,23], walls under out-of-plane loading [24,25,26,27], vaults, and arches [28,29,30,31,32]. In these systems, the bond problem arises between the matrix and the textile since it is not possible for the mortar with a certain grain size to penetrate well between the very thin filaments, which form the fiber bundle of the textile reinforcement. In these systems, the bond between the textile and the mortar is only in the form of the bond between the outer filaments of the bundle and the mortar around. As a result, failure in the composite system occurs in a so-called “telescopic” manner as a result of different deformations of the filaments in the outer environment and the filaments in the textile bundle [33]. The addition of short fibers into the mortar to improve the behavior of textile-reinforced cementitious composite systems was investigated in the literature [34,35]. As stated in these studies, with the addition of short fibers into the mortar, the stress value at which the first crack occurs under the tensile effects in the hybrid system increases, and the energy dissipation ability is significantly improved. The length of the effectively deforming region in the hybrid system also increases with the crack-bridging effect of short fibers. On the other hand, glass fiber reinforced mortar (also known as glass fiber reinforced concrete) is a composite material consisting of alkali-resistant glass fiber reinforcement, Portland cement, fine aggregate, water, and additives that is mainly used for producing structural elements such as façade and acoustic panels, melioration and drainage systems, ducts, as well as in municipal sewage systems, etc. [36]. The proposed study combines glass fiber reinforced mortar and basalt textile reinforcement to harness the favorable properties of each component to obtain a composite material that can be used for strengthening of existing structures. This includes crack resistance and the bridging effect of glass fiber reinforced mortar and the strength provided by the basalt mesh. Moreover, an innovative spraying method was incorporated into the production of composite material, which remarkably reduces required time and manpower.

In this study, the mechanical behaviors of sprayed glass fiber-reinforced mortar with and without basalt textile reinforcement, which were proven to be effective to enhance the stress–strain relationships of low-strength concrete members by external jacketing were investigated [12]. In the sprayed mortar, two different ratios of glass fiber were used as 3.5% and 5% by weight. In addition to compression tests, a total of 50 specimens were subjected to flexural and tensile strength tests, and the findings were interpreted.

## 2. Experimental Program

The tensile and flexural tests were carried out on the composite coupon specimens. The specimens were named as X-Y-Z where X was used to differentiate test type (T for tensile, and F for flexure), Y is used to denote composite configuration (%3.5 for the specimens that contain only 3.5% glass fiber by weight) (BG-1, BG-2, and BG-3 for the specimens that contain one, two, or three layers of basalt textile reinforcement in addition to 3.5% glass fiber) (%5 for the specimens that contain only 5% glass fiber by weight), and Z corresponds to specimen number. Six and four identical specimens were tested for each composite configuration in tensile and flexural tests, respectively.

### 2.1. Materials

In the study, the mortar used to produce the specimens consisted of CEM I 52.5 R white Portland cement, silica sand, metakaolin, latex additive, water, and superplasticizer. The chemical content of Portland cement in accordance with the ASTM C150 [37] standard was preferred. The standard chemical contents of the cements are given in Table 1. The gradation of fine aggregate was not given significance because it is believed that the distribution of silica-based fine aggregate has little impact on strength [38,39]. Some physical and mechanical properties of glass fibers and basalt textile reinforcement provided by the manufacturers are given in Table 2.

The basalt textile reinforcement is bidirectional, PVC-coated with an equal amount of basalt fibers, and then woven into an open grid. Mesh size of basalt textile reinforcement is 25 mm in two orthogonal directions (Figure 1).

### 2.2. Preparation of Composite Samples

The amounts of ingredients in the mixture were 900 kg of cement, 1000 kg of fine silica sand, 100 kg of metakaolin, 33 kg of latex additive, 320 L of water, and 2.4 kg superplasticizer for 1 m^3^. The compressive strength of the mortars was tested on a total of 12 cube specimens of 50 × 50 × 50 mm^3^ with only 3.5% and 5% glass fiber ratios. The specimens were tested according to ASTM C109/C109M [41] using a servo-hydraulic concrete press at the age of 28 days.

To produce coupon composite specimens, five 1500 mm × 1000 mm × 25 mm molds were created. First, basalt textile reinforcement was cut to the mold’s dimensions (Figure 2a). A spray gun was used to spray the first layer of glass fiber mortar into the mold (Figure 2b). The spraying thickness was checked at least at five different points (Figure 2c). The basalt textile reinforcement layer was placed into the mold and then covered with another layer of sprayed mortar. For specimens containing more than one basalt layer, the procedure was repeated until the final layer was reached and covered. The spraying process was performed continuously for the specimens without basalt textile reinforcement throughout the mold thickness. It should be noted that the thickness for all the composite configurations was 25 mm and basalt textile reinforcement layers were symmetrically placed throughout this thickness (Figure 3a,b). Panels were cured for seven days at laboratory conditions with the temperature of 21 ± 2 °C and 95 ± 5% relative humidity. Composite coupon samples were obtained by cutting the panels with the dimensions of 500 mm × 75 mm (Figure 3c,d). Hojdys and Krajewski [14] also obtained coupon samples of certain sizes from large panels. The coupon dimensions were selected considering the requirements of RILEM TC 232-TDT [42].

### 2.3. Tensile Test Procedure

At the building materials laboratory at Istanbul Technical University, composite samples were subjected to direct tensile test using an MTS Criterion Model 43 electromechanical universal tensile equipment.

Specimens were clamped between two rigid steel plates at the top and the bottom and rubber plates with a thickness of 0.5 mm were placed between the steel plates and specimens. Specimens were connected to the testing machine via a hinged connection so that the rotation in the plane of the specimen was allowed. Figure 4a schematically illustrates the experimental setup.

The tensile forces during the tensile tests were measured using integrated load cell, and the deformations were measured using two LVDTs (Linear Variable Differential Transducer) attached to the both sides of the test specimen (Figure 4b) and were recorded in a computer using a TML TDS-302 data logger. The tensile strain was computed by averaging the data from the two LVDTs. It should be noted that a 25-mm gap was left between the measurement frame where LVDTs were attached and the loading plates. Although approximately fifty tensile tests were conducted, only thirty tests (six specimens for each of the five configurations) in which a final crack formed in the measurement zone were considered for the evaluation of the results. Throughout the duration of the experiment, the loading rate was fixed at 0.5 mm/min. Tensile tests were performed as per RILEM TC 232-TDT [42].

According to the literature, the failure modes of specimens under direct tension [12,43,44] are generally categorized into three modes. Mode 1 is defined as clamp failure, Mode 2 as the cracking of the matrix, and Mode 3 as the rupture of the fiber in the matrix. Failure mode was defined according to the findings obtained. Tensile stress $\left(\sigma_{t}\right)$ was calculated using the formula $\sigma_{t}=f_{t}/A_{c}$ where *f_t_*: tensile force, *A_c_*: composite cross-sectional area. Tensile strain $\left(\varepsilon_{t}\right)$ was obtained as $\varepsilon_{t}=\Delta_{l}/l_{0}$ where Δ*_l_*: the average of the displacement values taken with LVDTs, *l*_0_: the measurement length of LVDTs (200 mm).

The modulus of elasticity of the specimens in the uncracked state was calculated using the slopes of the linear portions of the stress–strain relationship up to the first crack. The cracked modulus of elasticity (*E_cracked_*) was determined according to Arboleda et al. [13] using the following relation: $E_{cracked}=\left(0.9\sigma_{t}-0.6\sigma_{t}\right)/\left(\varepsilon @0.9\sigma_{t}-\varepsilon @0.6\sigma_{t}\right).$ In the equation, *σ_t_*: composite tensile strength; *ε@*0.9*σ_t_*: tensile strain value corresponding to 90% of the composite tensile strength; *ε@*0.6*σ_t_*: tensile strain value corresponding to 60% of the composite tensile strength.

### 2.4. Flexural Test Procedure

The flexural tests (four-point bending tests) were executed on the coupon specimens according to ASTM C 947 [45] using an MTS universal testing device. Load values were recorded with the help of the load cell of the device while displacement values were measured with two LVDTs and recorded in a computer environment using the TML TDS 302 data logger. The setup of the flexural test is given in Figure 5.

Flexural stress $\left(\sigma_{f}\right)$ and modulus of elasticity in flexure ($E_{f})$ were, respectively, calculated with $\sigma_{f}=M/W$ and $E_{f}=(27P_{y}L^{3})/(Y_{y}bd^{3})$ formulas. In the equation, *M*: maximum moment in the specimen, *W*: section modulus, *L*: loading distance between the supports, $P_{y}$ and *Y_y_*: measured force and displacement values where the load–displacement curve departs from linearity, *d*: average specimen depth.

## 3. Findings

### 3.1. Compressive Strength Test Findings

The average compressive strength of six specimens for the configuration with a ratio of 3.5% glass fiber was 43 MPa, while the average compressive strength of the configuration with a ratio of 5% glass fiber was approximately 46.1 MPa (Table 3). Similar to the result reported by Zhang et al. [4], the compressive strength of the mortar increased slightly as the glass fiber ratio in the mortar mixture increased. In their study investigating the effect of glass fiber addition on the compressive strength of fine aggregate concrete, the compressive strength values of 39.7 and 41.6 MPa were obtained for 18 mm length fiber mortar with 2% and 5% fiber ratios, respectively [4]. Similar results were obtained in studies conducted by Ghugal and Deshmukh [46], Hilles and Ziara [11], and Malek et al. [47].

### 3.2. Tensile and Flexural Tests of Composite with 3.5% Glass Fiber

The average tensile test results of six composite specimens with 3.5% glass fiber are given in Table 4. Accordingly, the average tensile strength of the specimens with 3.5% glass fiber was 5.98 MPa, while the tensile strain at maximum strength was 0.0061. The average uncracked modulus of elasticity of the specimens was calculated as 27,692 MPa. Since no significant hardening was detected after cracking, cracked modulus of elasticity values could not be calculated. The stress–strain relationships of the specimens with 3.5% glass fiber are given in Figure 6a, and the post-test images are given in Figure 6b.

When the maximum tensile strength was reached, there was a very abrupt drop in strength; therefore, only the portions up to the maximum strength value are shown in Figure 6a. During the tensile tests, the behavior was linear elastic until the occurrence of the first crack in the transverse direction; following the first crack, the slope of the tensile stress–strain curve reduced dramatically and became almost horizontal. After the cracking strength was exceeded, it was observed that numerous capillary cracks formed on the specimen. One of these cracks grew into a main crack, and after the development of this main crack, there was a sudden decrease in strength. With the expansion of the main crack, it was observed that the other hairline cracks closed due to the bridging of the short fibers around them and there was only a single fracture surface (Figure 6b). The specimens failed after the development of the main crack, when the short fibers surrounding this crack slipped from the mortar.

The average flexural strength of the four specimens was 15.76 MPa, and the average corresponding mid span displacement to the flexural strength was 9.4 mm. The flexural stress–displacement relationship varied linearly until the formation of the first crack in the region of maximum moment, and after the formation of the first crack, the slope decreased until the flexural strength. The average modulus of elasticity was calculated as 27,646 MPa for the specimens. The flexural stress–displacement relationship is given in Figure 7a. The average behavior of the four specimens is also shown in red on the graph. During the experiment, a single main crack in the transverse direction was observed on the specimen (Figure 7b).

### 3.3. Tensile and Flexural Tests of Composite with 5% Glass Fiber

The tensile test results, standard deviation, and coefficient of variation values for the composite configuration with 5% glass fiber are given in Table 5. The average tensile strength of specimens was 7.42 MPa, while the average tensile strain value corresponding to the maximum tensile strength was 0.0077. The average value of the uncracked modulus of elasticity was calculated as 26,916 MPa. The cracked modulus of elasticity values for 5% were not calculated because no significant hardening was observed after cracking as was the case for the composite configuration with 3.5% of the glass fiber. The behavior and mode of failure observed during the experiment were similar to those described above for specimens with the ratio of 3.5% glass fiber. Figure 8a shows the tensile stress–tensile strain relationships of specimens, while Figure 8b depicts the post-test views.

Figure 9a shows the flexural stress–displacement relationship. Similar to the composite specimens containing 3.5% glass fiber, a single major crack was observed in the specimen’s transverse direction during testing (Figure 9b). The average flexural strength of composite specimens containing 5% glass fiber was 17.13 MPa. The average specimen displacement corresponding to this strength was 9.1 mm. The average modulus of elasticity of the samples in the flexure was calculated as 27,514 MPa.

### 3.4. Tensile and Flexural Tests of Composites with 3.5% Glass Fiber and One Layer of Basalt Textile Reinforcement

The average tensile test results of six composite specimens containing 3.5% glass fiber and one layer of basalt textile reinforcement are given in Table 6. It can be seen that the specimens had an average tensile strength of 7.17 MPa and an average strain value of 0.0078 at their tensile strength. The cracked modulus of elasticity was 396 MPa and the uncracked modulus of elasticity was 26,463 MPa. Tensile stress–tensile strain relationships and close view of the final crack were depicted in Figure 10a and Figure 10b, respectively.

After the initial crack formation in the transverse direction, numerous capillary cracks appeared on the specimen during the experiments; one of these cracks then continued to develop and became the primary crack (Figure 10b). Short glass fibers bridging around these crack segments had the effect of bridging other hairline cracks after the development of the primary crack. In addition, it was observed that the cracks formed in the line of the transverse basalt textile bundles during the test. The specimens finally failed due to the rupture of the basalt textile reinforcement (Figure 10b). As it was observed in composite specimens without basalt textile reinforcement, the short glass fibers surrounding the main crack cross-section did not rupture but slipped out of the mortar.

Figure 11a shows the flexural stress–displacement relationships. On the figure, the curve representing the average behavior of the four specimens is also shown in red. During the testing of composite specimens containing 3.5% glass fiber and one layer of basalt textile reinforcement, a single major transverse crack was observed (Figure 11b). The specimens’ average flexural strength was 16.68 MPa, and the corresponding displacement to the flexural strength was 11.3 mm. The average modulus of elasticity of the specimens in flexure was 27,640 MPa.

### 3.5. Tensile and Flexural Tests of Composite with 3.5% Glass Fiber and Two Layers of Basalt Textile Reinforcement

The mean stress, standard deviation, and coefficient of variation for composite specimens containing 3.5% glass fiber and two layers of basalt textile reinforcement are provided in Table 7. The average tensile strength of the specimens was determined as 8.16 MPa, while the average tensile strain value corresponding to the maximum tensile strength was determined as 0.0074. The average uncracked modulus of elasticity was calculated as 26,066 MPa and the average cracked modulus of elasticity was calculated as 501 MPa. Tensile stress and tensile strain relationships and view of specimens after the test are depicted in Figure 12. Damage progression was similar with composite specimens containing 3.5% glass fiber and one layer of basalt textile reinforcement.

Figure 13a shows the relationships between flexural stress and displacement. A single major crack was observed in the specimens’ transverse direction. As the load increased, this crack moved towards the compression zone and began to propagate in the longitudinal direction at the level of the basalt textile layer near the top surface of the specimen due to the loss of the bond between this textile layer and the mortar. This was more pronounced in F-BG-2-3 and F-BG-2-4 specimens than in F-BG-2-1 and F-BG-2-2 specimens (Figure 13b). It was determined that the average flexural strength of these specimens was 20.09 MPa and the average displacement value corresponding to this strength was 12.1 mm. The calculated average flexural modulus of elasticity for the specimens was 28,643 MPa.

### 3.6. Tensile and Flexural Tests of Composite with 3.5% Glass Fiber and Three Layer of Basalt Textile Reinforcement

The mean stress, standard deviation, and coefficient of variation values for composite specimens containing 3.5% glass fiber and three layers of basalt textile reinforcement are given in Table 8. The average tensile strength was determined as 9.7 MPa, while the average tensile strain value corresponding to the maximum tensile strength was determined as 0.0091. The average uncracked modulus of elasticity was calculated as 27,404 MPa, and the average cracked modulus of elasticity was calculated as 554 MPa. The tensile stress and tensile strain relationships obtained at the end of the tensile tests on composite specimens are given in Figure 14a. The specimens failed due to rupture of the basalt textile reinforcement or delamination (separation between layers) (Figure 14b), unlike the configurations with one and two layers of basalt textile reinforcement in addition to 3.5% glass fiber. The short glass fibers around the main crack cross-section did not rupture but slipped from the mortar, as in the composite specimens with glass fibers only, as outlined above.

In the composite specimens containing 3.5% glass fiber and three layers of basalt textile reinforcement, a single main crack was observed in the transverse direction of the specimen during flexural tests (Figure 15b). This crack progressed towards the compression zone as the load increased and started to progress in the transverse direction at the level of the basalt textile layer closest to the bottom of the specimen in F-3.5%-3-3 and F-3.5%-3-4 specimens because of the loss of the bond between this textile layer and mortar. This was more evident in F-BG-3-3 and F-BG-3-4 specimens than in F-BG-3-1 and F-BG-3-2 specimens, and the flexural strengths of these specimens were slightly lower than the other specimens (Figure 15a,b). The average flexural strength of the tested specimens was 20.64 MPa, and the average mid-sample displacement corresponding to this strength was 13 mm. The average modulus of elasticity of the specimens in flexure was calculated as 27,852 MPa.

## 4. Discussion

### 4.1. Tensile Behavior

Figure 16 shows the average tensile stress–tensile strain relationships of tested composite systems. For the composite systems that contain only short fibers, the average stress value at which the first crack occurred during tensile tests in the composite system containing 5% glass fibers (5.41 MPa) was greater than the that of the stress value for the composite system containing 3.5% short glass fibers (4.73 MPa) (Figure 16). Kwan and Chu [48] also highlighted the increase in the first crack stress with the increase in fiber amount. The enhancement ratios for the tensile strength and tensile strain with the increase of fiber amount from 3.5% to 5% were, respectively, calculated as 24% and 26%. However, it can be stated that the improvement in behavior was modest in comparison to the increase in fiber. The failure of these two configurations was the similar and due to slip of the short fibers. Considering all the facts summarized above, the configuration of 3.5% short glass fibers was preferred as the matrix for configurations with basalt textile reinforcement.

The highest tensile strength and strain values were obtained for the composite configuration containing 3.5% glass fiber and three layers of basalt textile reinforcement (Figure 16). As the number of basalt textile reinforcement layers in 3.5% glass fiber specimens increased, the tensile strength and tensile strain values corresponding to the tensile strength increased. This is because the tensile behavior of the basalt textile reinforcement is better than that of the glass fiber mortar. The enhancement rates in tensile strength were 20%, 44%, and 62% for composite configurations with one, two, and three layers of basalt textile reinforcement, respectively, while the increase rates for the ultimate tensile strain values corresponding to the tensile strength were 28%, 21%, and 49% for the same configurations. The improvement of tensile strength with the increase in the number of textile reinforcement layers placed in the fiber mortar was also reported by Zhu et al. [49]. Furthermore, no significant slip between the basalt textile reinforcement and mortar was observed during tensile tests. For the composite configurations containing 3.5% glass fibers and basalt textile reinforcement, the observed failure was generally due to slipping of glass fibers around the main and the rupture of basalt textile reinforcement at the crack section. This is consistent with the failure mode reported by Zhu et al. [49]. The main crack occurred in line with the transverse textile bundles, perpendicular to the tensile direction. Textile bundles perpendicular to the tensile direction cause weakening of the section in the loading direction as emphasized by Valeri et al. [50]. However, for the configuration containing 3.5% glass fiber and three layers of basalt textile reinforcement, a slight delamination was observed in some of the tested specimens. The occurrence of delamination as the number of textile reinforcements in the composite increases has also been reported by Donnini and Corinaldesi [51] and Younis and Ebead [52].

Presence of basalt textile reinforcement in the sprayed glass fiber spray mortar did not significantly affect the uncracked modulus of elasticity and the cracking stress. The main effect of basalt textile reinforcement was clearly seen in the tensile stress–tensile strain relationship after cracking (Figure 16). As the number of basalt textile reinforcement layers increased, the slope of the hardening part of the curve after cracking clearly increased. A similar trend was also reported by Rambo et al. [53] and Zhu et al. [49]. The numerical values of the slope of this section were documented using the composite system cracked modulus of elasticity. The average cracked section modulus of elasticity values were calculated as 396 MPa, 501 MPa, and 554 MPa for composite systems containing one, two, and three layers of basalt textile reinforcement, respectively. The relevant value could not be calculated for the configuration containing only 3.5% by weight glass fiber, as no significant hardening occurred after cracking.

### 4.2. Flexural Behavior

Williams Portal [54] stated the behavior of the textile reinforced mortar composite specimens (without short fibers) under flexure can be considered in three different stages. In the first stage, the composite has not yet cracked, and stiffness is a function of the matrix material. After the cracking strength of the matrix is reached and the first crack is formed, tensile stresses begin to develop in the textile reinforcement. In the second stage, the number of cracks increases with increasing load, and the load–displacement relationship fluctuates without a significant reduction in load until the crack formation stabilizes and the mechanical properties of the textile reinforcement are effective. In the final stage, the strength loss of the composite occurs under flexural effects when the textile reinforcement reaches the strength loss or peels off from the matrix. The flexural stress–displacement relationships were linear for all the tested composite configurations until the occurrence of the first crack. The slopes of this part were calculated as 27,646 MPa, 27,640 MPa, 28,643 MPa, and 27,852 MPa for the F-3.5%, F-BG-1, F-BG-2, and F-BG-3 configurations. These values are very close to each other, indicating that the textile reinforcement (basalt) does not have a significant effect on the behavior until the first crack, as stated by Williams Portal [54]. However, the multiple crack zone reported by Williams Portal [54] was not observed in the tested samples. Contrary to this situation, the load (stress) values tended to increase with the increasing strain immediately after the crack formation (Figure 11a, Figure 13a and Figure 15a). This can be attributed to the uninterrupted stress transfer around the crack with the short glass fibers acting as a bridge between the cracks. Kong et al. [55] also reported a similar situation. After the initial crack formation, a significant decrease in the slope of the flexural stress–displacement curve occurred, with an increasing stress trend with increasing displacement until failure for all tested configurations. This behavior is consistent with the flexural stress–displacement relationships reported in similar studies in the literature (Kong et al. [55]; Wang et al. [56]).

Comparing the behavior of composite specimens without basalt textile reinforcement and those containing only 3.5% and 5% chopped glass fiber, it was found that the first crack occurred at higher stress level in the composite system containing 5% glass fiber due to the higher proportion of fibers resisting tensile stresses. The first crack occurred at a flexural stress value of approximately 6.75 MPa in composite specimens containing 3.5% glass fiber, while it occurred at approximately 7.60 MPa in composite specimens containing 5% glass fiber (Figure 17). The flexural modulus of elasticity was calculated to be 27,646 MPa for composite specimens containing 3.5% short glass fiber and 27,514 MPa for composite specimens containing 5% short glass fiber. It was also emphasized by Wafa [57] that the amount of fiber did not significantly affect the modulus of elasticity of the composite system. As it was the case in the tensile tests, the enhancement with the increase in the short fiber ratio was found to be modest.

## 5. Conclusions

In this study, the tensile and flexural behaviors of sprayed glass fiber reinforced mortar with/without basalt textile reinforcement were examined through an experimental campaign. The main findings of the study are summarized below.

The tensile and flexural capacities, and the first cracking strength marginally improved when the ratio of chopped glass fibers increase from 3.5% to 5% for the composite configurations without basalt textile reinforcement. As the number of basalt textile reinforcement layers increased in the mortar that contains 3.5% glass fiber, the tensile strength and tensile strain values corresponding to the tensile strength improved. The highest increase ratios were obtained for the composite configuration with three layers of basalt textile reinforcement.

Presence of basalt textile reinforcement in the sprayed glass fiber spray mortar did not significantly affect the uncracked modulus of elasticity and the cracking stress in tension. The main effect of basalt textile reinforcement was clearly seen in the tensile stress–tensile strain relationship after cracking. As the number of basalt textile reinforcement layers increased, the slope of the hardening part of the curve after cracking clearly increased.

Unlike tensile tests, no significant difference was observed in flexural behavior for the composite configurations with two or three layers of basalt textile reinforcement. The failure mode in flexure turned out to be delamination when the number of basalt textile layers was more than two. The direct tensile strength can roughly be assumed to be half of the flexural strength for composites with/without basalt textile reinforcement.

The tensile and flexural tests showed that utilizing of sprayed glass fiber reinforced mortar and basalt textile reinforcement is a promising technique to obtain cement-based composite systems. However, the results presented in this study were based on a limited number of test specimens. Therefore, further studies are required to better determine the mechanical behavior of the proposed composite systems.

## Figures and Tables

**Figure 1 materials-16-04251-f001:**
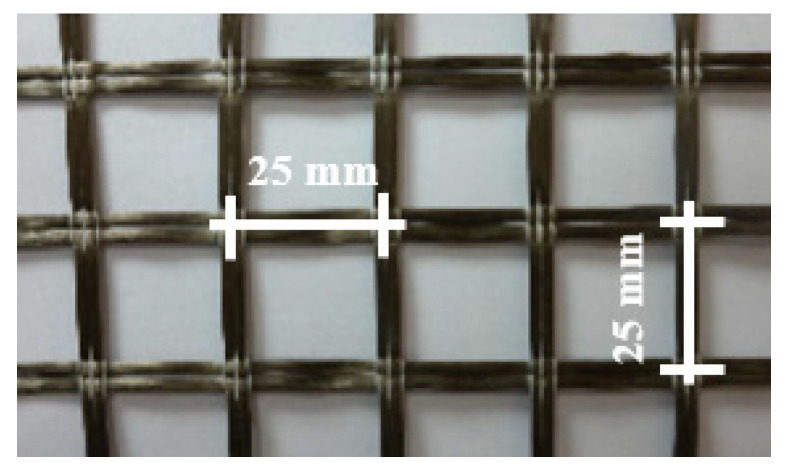
Basalt textile reinforcement.

**Figure 2 materials-16-04251-f002:**
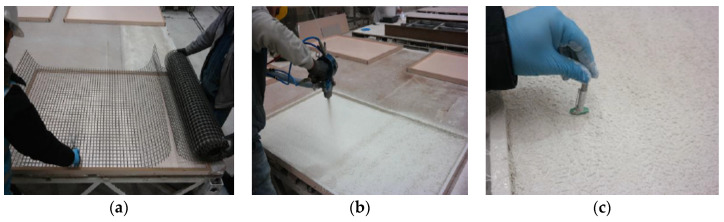
(**a**) Basalt textile reinforcement. (**b**) Spraying process. (**c**) Checking the thickness.

**Figure 3 materials-16-04251-f003:**
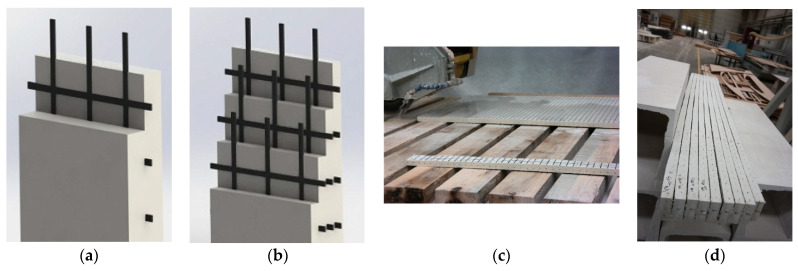
Arrangement of basalt textile reinforcement inside the composite thickness for (**a**) one and (**b**) three layers, (**c**,**d**) cutting of coupons from panels.

**Figure 4 materials-16-04251-f004:**
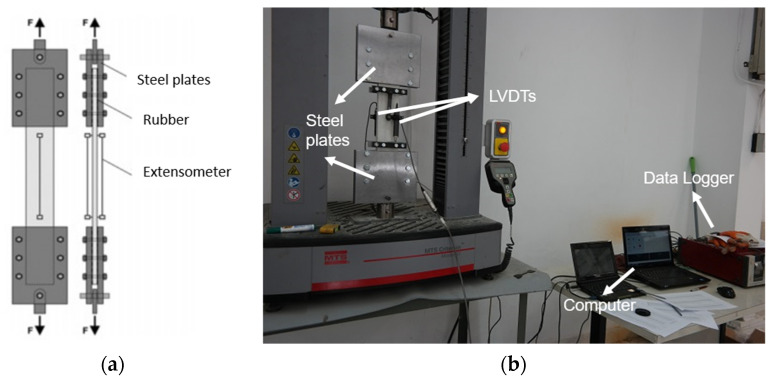
(**a**) Tensile test setup (schematical) [42]. (**b**) Tensile test setup.

**Figure 5 materials-16-04251-f005:**
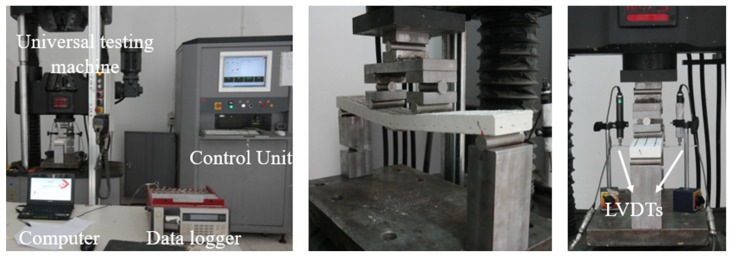
Flexural test setup.

**Figure 6 materials-16-04251-f006:**
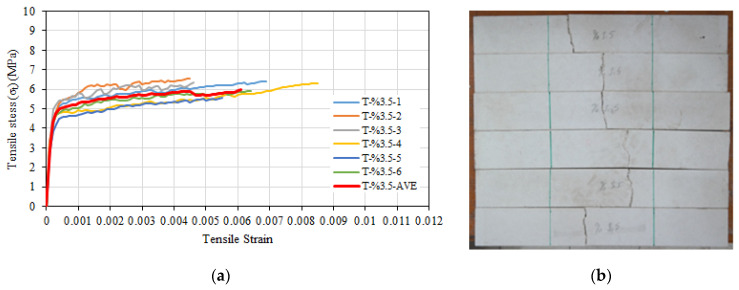
(**a**) Tensile stress–tensile strain relationships of T-%3.5 series. (**b**) Post-test view.

**Figure 7 materials-16-04251-f007:**
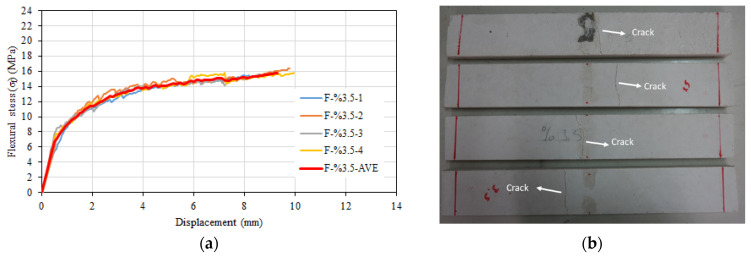
(**a**) Flexural stress–displacement relationship of F-%3.5 series. (**b**) Post-test view.

**Figure 8 materials-16-04251-f008:**
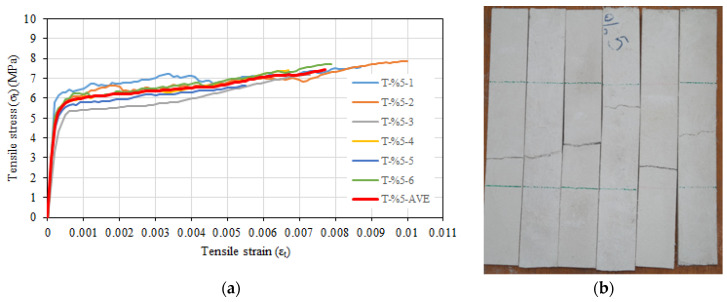
(**a**) Tensile stress–tensile strain relationships of T-%5 series. (**b**) Post-test view.

**Figure 9 materials-16-04251-f009:**
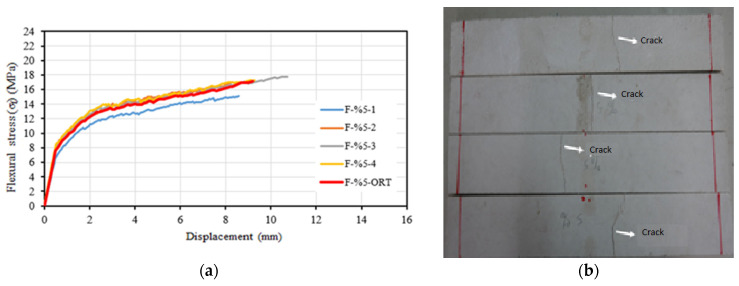
(**a**) Flexural stress–displacement relationships of F-%5 series. (**b**) Post-test view.

**Figure 10 materials-16-04251-f010:**
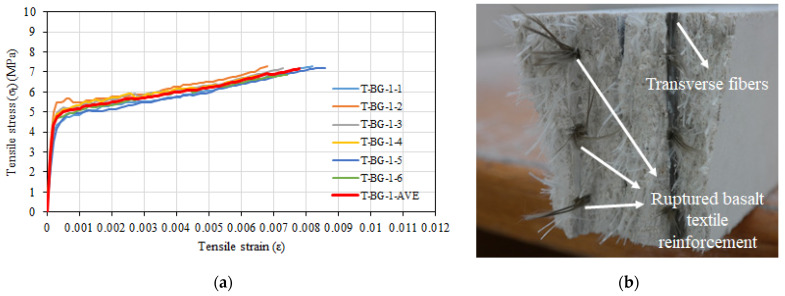
(**a**) Tensile stress–tensile strain relationships of T-BG-1 series. (**b**) View of failure surface.

**Figure 11 materials-16-04251-f011:**
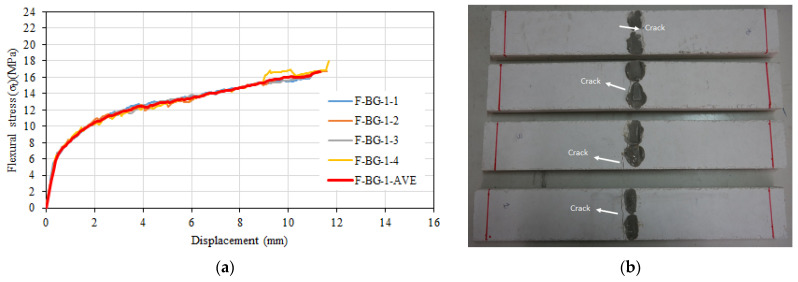
(**a**) Flexural stress–displacement relationships of F-BG-1 series. (**b**) Post-test view.

**Figure 12 materials-16-04251-f012:**
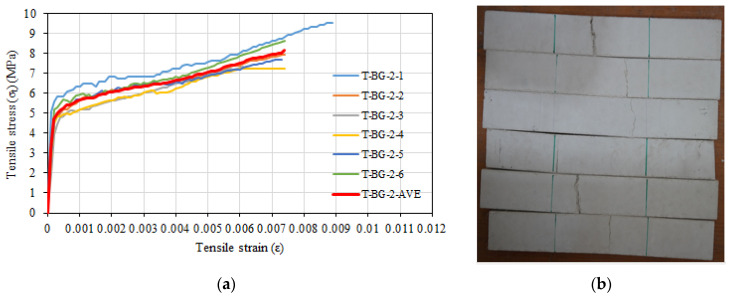
(**a**) Tensile stress–tensile strain relationships of T-BG-2 series. (**b**) View of specimens after test.

**Figure 13 materials-16-04251-f013:**
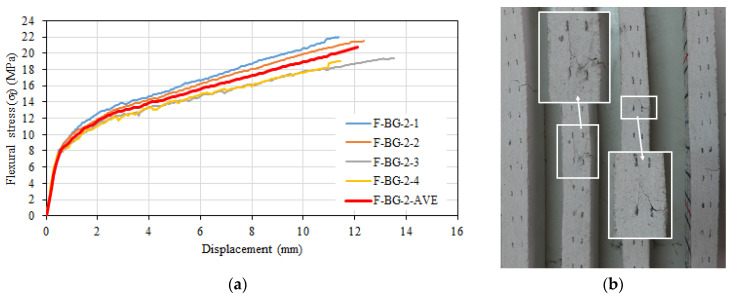
(**a**) Flexural stress–displacement relationships of F-BG-2 series. (**b**) Side view after test.

**Figure 14 materials-16-04251-f014:**
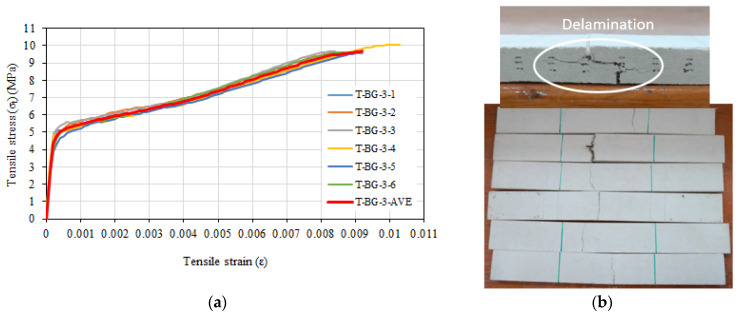
(**a**) Tensile stress–tensile strain relationships of T-BG-3 series. (**b**) Top and side view.

**Figure 15 materials-16-04251-f015:**
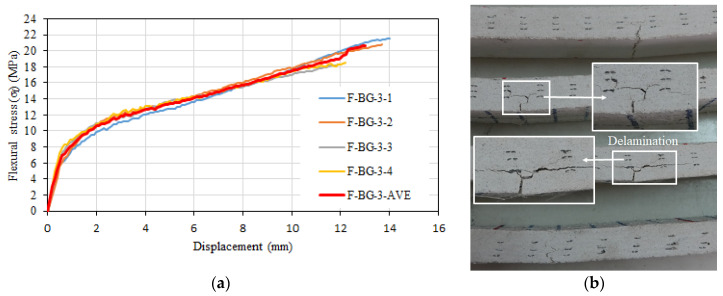
(**a**) Flexural stress–displacement relationships of F-BG-3 series. (**b**) Side view of the specimens after test.

**Figure 16 materials-16-04251-f016:**
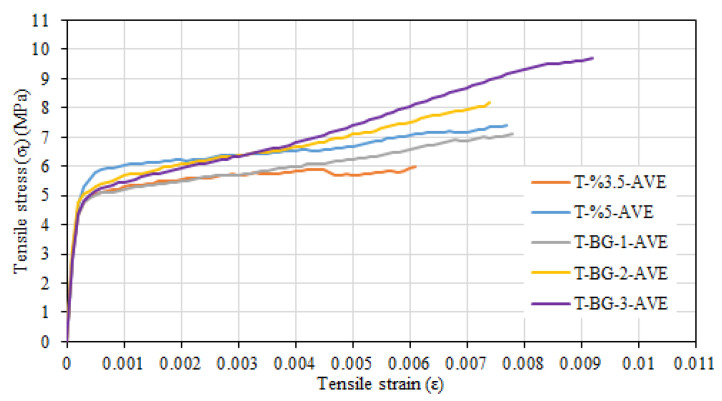
Average tensile stress–tensile strain relationships for tested composite configurations.

**Figure 17 materials-16-04251-f017:**
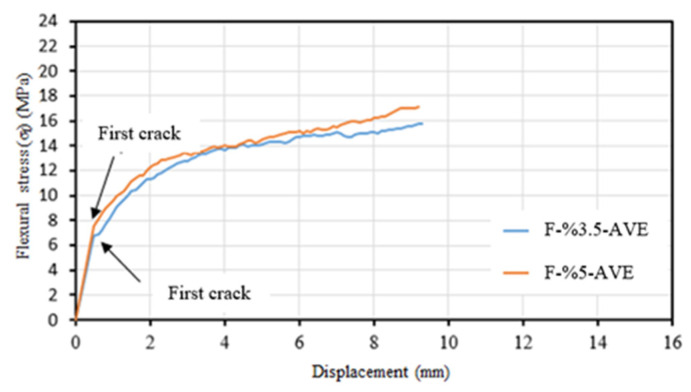
Average flexural stress–displacement curves for F-%3.5 and F-%5 series.

**Table 1 materials-16-04251-t001:** Chemical contents of cements.

Constituent/Property	White Portland Cement	Portland Cement	Deák et al. wt% [40]
SiO_2_ (%)	17.95	19.34	42.43–55.69
Al_2_O_3_ (%)	2.98	4.55	14.21–17.97
Fe_2_O_3_ (%)	0.21	2.77	10.80–11.68
CaO (%)	59.40	62.43	7.43–8.88
MgO (%)	2.87	2.61	4.06–9.45
SO_3_ (%)	3.09	2.89	<5
Na_2_O (%)	0.43	0.09	<5
K_2_O (%)	0.36	0.74	<5
Loss on ignition (%)	11.60	2.09	-

**Table 2 materials-16-04251-t002:** Properties of glass and basalt textile reinforcement.

	Glass Fiber	Basalt Textile Reinforcement
Fiber length	About 24 mm	-
Filament Diameter	14 µm	8–9 µm
Density	2.8 g/cm^3^	2.62–2.65 g/cm^3^
Modulus of Elasticity	74 GPa	-
Tensile strength	1500 MPa	95 kN/m
Softening point	860 °C	-
ZrO_2_ content	17%	-
Surface weight	-	303 g/m^2^
Breaking strain	2%	5%
Mesh Size	-	25 mm

**Table 3 materials-16-04251-t003:** Compressive strength, MPa.

Fiber Ratio in the Mortar (% by Weight)	Average Compressive Strength (MPa)		
	f_c_ (MPa)	Std. Dev.	CoV
3.5	43.0	2.1	4.9
5.0	46.1	1.7	3.7

Std. Dev.: Standard Deviation, CoV: Coefficient of variation.

**Table 4 materials-16-04251-t004:** Tensile test results of specimens containing 3.5% glass fiber.

	*σ_t_* (MPa)	*ε_t_*	*E_uncracked_* (MPa)	*E_cracked_* (MPa)
Mean	5.98	0.0061	27,692	-
Standard deviation	0.34	0.0015	2404	-
Coefficient of variation (%)	5.69	24.6	8.68	-

**Table 5 materials-16-04251-t005:** Tensile test results of specimens containing 5% glass fiber.

	*σ_t_* (MPa)	*ε_t_*	*E_uncracked_* (MPa)	*E_cracked_* (MPa)
Mean	7.42	0.0077	26,916	-
Standard deviation	0.43	0.0016	5624	-
Coefficient of variation (%)	5.75	20.39	20.89	-

**Table 6 materials-16-04251-t006:** Tensile test results of specimens containing 3.5% glass fiber and one layer of basalt textile reinforcement.

	*σ_t_* (MPa)	*ε_t_*	*E_unracked_* (MPa)	*E_cracked_* (MPa)
Mean	7.17	0.0078	26,463	396
Standard deviation	0.21	0.0007	4092	31
Coefficient of Variation (%)	2.93	8.97	15.5	7.8

**Table 7 materials-16-04251-t007:** Tensile test results of specimens containing 3.5% glass fiber and two layers of basalt textile reinforcement.

	*σ_t_* (MPa)	*ε_t_*	*E_uncracked_* (MPa)	*E_cracked_* (MPa)
Mean	8.16	0.0074	26,066	501
Standard deviation	0.88	0.0009	3205	80
Coefficient of Variation (%)	10.8	12.1	12.3	16

**Table 8 materials-16-04251-t008:** Tensile test results of specimens containing 3.5% glass fiber and three layers of basalt textile reinforcement.

	*σ_t_* (MPa)	*ε_t_*	*E_uncracked_* (MPa)	*E_cracked_* (MPa)
Mean	9.7	0.0091	27,404	554
Standard deviation	0.20	0.0006	2711	37
Coefficient of Variation (%)	2.06	6.59	9.9	6.7

## Data Availability

The data presented in this study are available on request from the first author.

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
