# Peer review of "Tensile and Flexural Behaviors of Basalt Textile Reinforced Sprayed Glass Fiber Mortar Composites"

_materials, 2023, doi:10.3390/ma16124251_

Round 1

Reviewer 1 Report

Look at the attachment.

it is fine

Author Response

The detailed responses to the comments are given below:

Review 1

This work investigated the mechanical behavior of basalt textile reinforced sprayed glass fiber mortar. The contents should be polished under a major revision. Here are my comments.

Authors wish to thank the reviewer for very valuable comments. The relevant changes are shown in the revised manuscript. We hope the changes satisfies the reviewer. The language of the manuscript and interpretations in the discussion section were also polished during the revision.

Comment 1: There is a lack of “of” in the title.

Response to comment 1: Many thanks for the correction. “OF” is added in the title (Line 2).

Comment 2: Please indicate the contribution and significance of your work in the abstract and introduction part.

Response to comment 2: Many thanks for the suggestion. Significance of the proposed research is added in the introduction (Lines 56-79) in detail and briefly in abstract (Lines 9-13). 

Comment 3: The contents of introduction are not satisfactory. There is no introduction to sprayed glass-reinforced mortar.

Response to comment 3: An introduction to glass-fiber reinforced mortar is added (Lines 69-74).

Comment 4: Usually, three different reinforcement ratios will be adopted to investigation the influence of fiber contents. In this work, only two ratios of glass fibers are adopted.

Response to comment 4: We appreciate for the comment. Considering the results of previous experience on sprayed glass-fiber reinforced mortar (unfortunately they are not published yet) and budget constraints, we decided to adopt two different fiber ratios -namely 3.5% and 5%- in the mortar. We should emphasize that the number of tested specimens are considerably high and we preferred to increase the number of tested specimens rather than increasing the number of configurations to eliminate the possible variations of the test results.

Comment 5: What is the weight ratio of basalt textile reinforcement?

Response to comment 5: Since the basalt textile reinforcement is continuous along the height of the specimen and independent of the mixture of mortar, we preferred to give the number of layers in the manuscript. However, the weight ratios of the basalt textile reinforcement was approximately 0.006, 0.012, and 0.018 for the composite configurations with one layer ,two layers, and three layers, respectively. If the reviewer suggests adding the corresponding weight ratios, we will be happy to add them in the manuscript.

Comment 6: Some notations are not correct, such as the average of the displacement values in line 156. Please check the whole manuscript.

Response to comment 6: We appreciate for the correction. Whole text of the manuscript was checked and such errors were corrected.

Comment 7: For each test, how many samples do you prepare?

Response to comment 7: For each configuration; six specimens were tested in tensile tests while four specimens were tested in flexural tests (Lines 95-96).

Comment 8: In figure 6(a), how is the average strain-stress curve obtained? I mean, for different specimen, they crack at different strains. Then, how do you draw the average curve.

Response to comment 8: To obtain average curve, we calculated the averages of corresponding stress values for the same strain up to the average ultimate strain value of the composite configuration. This method enables to draw the average curve between two or more curves.

 Comment 9:  Some new and fiber-related studies, such as Influences of MgO and PVA fiber on the abrasion and cracking resistance, pore structure and fractal features of hydraulic concrete; and Comparison of fly ash, PVA fiber, MgO and shrinkage-reducing admixture on the frost resistance of face slab concrete via pore structural and fractal analysis could improve the work.

Response to comment 9: We appreciate for the precious comment. We will consider the very valuable suggestion in our future research.

Comment 10:  Only the 3.5% glass fiber reinforced mortar were incorporated with extra basalt textile reinforcements. Why not 5%?

 Response to comment 9: The enhancement in the tensile and flexural behavior with the increase of glass fiber amount from 3.5% to 5% was found to be modest with respect to the increase in the fiber ratio (L381-382 and L457-458). Accordingly, 3.5% glass fiber reinforced mortar was preferred for the composite configurations with basalt textile reinforcement. 

Reviewer 2 Report

The article received for review, entitled: “Tensile and Flexural Behaviors Basalt Textile Reinforced Sprayed Glass Fiber Mortar Composites” is very relevant to the wider construction industry.

The article begins with a literature review, which is written correctly in terms of content. It contains interesting information relevant to the topic of the article. The literature list is relevant and thematically well chosen for this part of the article. The literature list can be considered very comprehensive, as it contains 52 items. Most of the literature items are from the last several years. This may indicate that the topic taken up by the authors is relevant and up-to-date.

Then, in Chapter 2, the authors present the research programme. The raw materials used to make the research samples are described in great detail. The research methods are described and the laboratory equipment used is presented. I found no factual errors in this section. In Section 3, the authors present the results of the tests obtained. The results are also presented correctly. The included graphs are clear and well prepared. The analysis of the obtained results is carried out correctly and in a way that is understandable for the reader. In Section 4, the authors have included a discussion, which is very interesting. I have no substantive comments on this point either.

The conclusions from the research (summary), which are included at the end of the article, are also correctly formulated. The discussion presented is also done factually correctly and shows well the results and conclusions of the research drawn by the authors. I have no substantive comments on them. I believe that the topic addressed by the authors is very relevant to structural engineering. There is no information in this summary as to whether the authors have used their experience in construction practice? If so, please present the conclusions of such practical work. And if not, I hope that the authors will continue to address this topic and develop it further. It is important that the results of this research can be implemented in practice as soon as possible.

In conclusion, I think that the article is generally written correctly and does not require any changes. I think that it can be published in an international journal such as Materials without significant amendments.

Author Response

Comment: The article received for review, entitled: “Tensile and Flexural Behaviors Basalt Textile Reinforced Sprayed Glass Fiber Mortar Composites” is very relevant to the wider construction industry.

The article begins with a literature review, which is written correctly in terms of content. It contains interesting information relevant to the topic of the article. The literature list is relevant and thematically well chosen for this part of the article. The literature list can be considered very comprehensive, as it contains 52 items. Most of the literature items are from the last several years. This may indicate that the topic taken up by the authors is relevant and up-to-date.

Then, in Chapter 2, the authors present the research programme. The raw materials used to make the research samples are described in great detail. The research methods are described and the laboratory equipment used is presented. I found no factual errors in this section. In Section 3, the authors present the results of the tests obtained. The results are also presented correctly. The included graphs are clear and well prepared. The analysis of the obtained results is carried out correctly and in a way that is understandable for the reader. In Section 4, the authors have included a discussion, which is very interesting. I have no substantive comments on this point either.

The conclusions from the research (summary), which are included at the end of the article, are also correctly formulated. The discussion presented is also done factually correctly and shows well the results and conclusions of the research drawn by the authors. I have no substantive comments on them. I believe that the topic addressed by the authors is very relevant to structural engineering. There is no information in this summary as to whether the authors have used their experience in construction practice? If so, please present the conclusions of such practical work. And if not, I hope that the authors will continue to address this topic and develop it further. It is important that the results of this research can be implemented in practice as soon as possible.

In conclusion, I think that the article is generally written correctly and does not require any changes. I think that it can be published in an international journal such as Materials without significant amendments.

Response: The authors wish to thank the reviewer for comprehensive comments and great interest. As pointed out by the reviewer, this kind of composite material was used and showed to be effective for the retrofit of low-strength concrete columns by external jacketing. This is emphasized in the revised manuscript (L81-84). We should add that the language and the discussion/conclision sections of the manuscript are further polished during the revision.